# Polyphenol-Rich Liupao Tea Extract Prevents High-Fat Diet-Induced MAFLD by Modulating the Gut Microbiota

**DOI:** 10.3390/nu14224930

**Published:** 2022-11-21

**Authors:** Waijiao Tang, Mengfei Yuan, Zewen Li, Qi Lin, Yan Zhen, Zhuang Li, Hongwei Zhou, Fangbo Xia

**Affiliations:** 1Microbiome Medicine Center, Department of Laboratory Medicine, Zhujiang Hospital, Southern Medical University, Guangzhou 510282, China; 2Shenzhen Stomatology Hospital (Pingshan), Southern Medical University, Shenzhen 518000, China

**Keywords:** MAFLD, gut microbiota, polyphenol-rich Liupao tea extract, fecal microbiome transplantation, LPS-TLR4-MyD88 signaling pathway

## Abstract

The modulation of gut microbiota dysbiosis might regulate the progression of metabolic-associated fatty liver disease (MAFLD). Here, we found that polyphenol-rich Liupao tea extract (PLE) prevents high-fat diet (HFD)-induced MAFLD in ApoE^−/−^ male mice accompanied by protection of the intestinal barrier and downregulation of lipopolysaccharide (LPS)-related Toll-like receptor 4 (TLR4)-myeloid differentiation primary response 88 (MyD88) signaling in the liver. Fecal microbiome transplantation (FMT) from PLE-and-HFD-treated mice delayed MAFLD development significantly compared with FMT from HFD-treated mice. In this case, 16S rRNA gene sequencing revealed that *Rikenellaceae* and *Odoribacter* were significantly enriched and that *Helicobacter* was significantly decreased in not only the HFD+PLE group but also the HFD+PLE-FMT group. Furthermore, the level of 3-sulfodeoxycholic acid was significantly decreased in the HFD+PLE-FMT group compared with the HFD-FMT group. In conclusion, our data demonstrate that PLE could modulate the MAFLD phenotype in mice and that this effect is partly mediated through modulation of the gut microbiota.

## 1. Introduction

The prevalence of metabolic-associated fatty liver disease (MAFLD), which is also known as nonalcoholic fatty liver disease fatty liver disease (NAFLD), is increasing worldwide. MAFLD comprises a spectrum from simple steatosis or MAFLD to nonalcoholic steatohepatitis, which is defined histologically as hepatic inflammation in addition to hepatic steatosis [1]. Although the exact sequence of events in the development of MAFLD remains not entirely clear, gut microbiota dysbiosis has been detected frequently in patients with MAFLD [2]. In addition, germ-free mice receiving fecal microbiome transplantation (FMT) from MAFLD patients develop hepatic macrovascular steatosis, indicating a causal relationship between the gut microbiota and the development of MAFLD [3]. Therefore, targeting the gut microbiota and related methods, such as untargeted approaches, including probiotics and fecal microbiota transplantation (FMT), and precision microbiome-centered therapies, including engineered bacteria, postbiotics and phages, might be a novel therapeutic approach for the treatment of MAFLD [4,5,6,7,8]. Among these, tea polyphenols as probiotics have been found to be associated with a low prevalence of MAFLD [9], which aroused our interest, and our preliminary research also suggested that tea consumption would be an important method for regulating the flora [10].

Liupao tea, which is prepared from *Camellia sinensis* var. sinensis, is a dark tea that has been prevalent in many countries for at least 1500 years. Chemical analyses have shown that Liupao tea is enriched in various secondary metabolites, such as polyphenols, which are important natural compounds that exert antioxidant, anti-inflammatory, and hepatoprotective effects [11,12]. Moreover, tea polyphenols reportedly exert their anti-inflammatory effects through regularization of the gut microbiota [13]. Therefore, we hypothesized that polyphenol-rich Liupao tea extract (PLE) might beneficially affect MAFLD by inhibiting liver inflammation, which may be mechanistically linked to the gut microbiome.

We evaluated the therapeutic potential of PLE in MAFLD in ApoE^−/−^ mice receiving a high-fat diet (HFD), which is a widely used animal model of MAFLD induced by HFD [14] and investigated the causal role of the gut microbiota in the effects of PLE on MAFLD by FMT.

## 2. Materials and Methods

### 2.1. Preparation of the PLE Extract

Liupao tea samples were provided by Guangxi Wuzhou Jinyi Liubao Tea Industry Co., Ltd., Wuzhou, China with the manufacture date of 8 March 2020. The Liupao tea samples (800 g) were ground into powder and subsequently extracted with 8000 mL of boiling water twice for 30 min each. The percolates were combined and concentrated in a rotary vacuum to 2000 mL. Subsequently, the water solution of Liupao tea extract was subjected to XAD-16 resin column separation, rinsed with 5 bed volumes of water, and eluted with pure ethanol. The polyphenol-rich PLE was dried using a rotary evaporator.

### 2.2. Determination of the PLE Composition

For chemical analysis, a 2-μL aliquot of PLE solution (0.5 mg/mL) was separated with a Waters ACQUITY HSS T3 C18 column (100 mm × 2.1 mm i.d., 1.7 μm) at 40 °C using a UPLC ExionLC AD system (SCIEX, MA). The mobile phase, which consisted of 0.1% formic acid-water (phase A) and acetonitrile (phase B), was pumped at a flow rate of 0.3 mL/min under the following program: isocratic 2% B (0–1 min), 2–70% B (1–5 min), 70–90% B (5–9 min), 90–100% B (9–9.5 min), isocratic 100% A for 2.5 min and then back to 3% B for 3 min. The QTOF-MS system X500R (SCIEX, MA), equipped with an electrospray ionization (ESI) turbo V source, was used to analyze the chemical profiles in the negative ion mode. The mass range was set to m/z 100 to 1200. The parameters were set as follows: source voltage, −4500 V, and decluttering potential (DP), −80 V. The IDA function was used for collecting TOF-MS and TOF-MS/MS data. The IDA settings were as follows: charge monitoring to exclude multiply charged ions and isotopes, dynamic background subtraction (DBS), source temperature, 550 °C; curtain gas flow, 30 psi, ion source gas 1, 50 psi, and ion source gas 2, 50 psi. To obtain accurate mass measurements, a calibrant delivery system (CDS) was used to maintain the mass accuracy.

### 2.3. Animals and PLE Supplementation

According to the report from Huang et al., MAFLD is more highly associated with males [15] because six-week-old male ApoE^−/−^ mice were randomly distributed into groups receiving a normal-fat diet (10% of energy from fat, normal diet, Guangdong Medical Laboratory Animal Center Co., Ltd., Guangzhou, China) or a high-fat diet (42% of energy from fat, 0.8% cholesterol added to the Western diet, Guangdong Medical Laboratory Animal Center Co., Ltd., Guangzhou, China) with or without supplementation of 150 mg/kg PLE suspended in sterilized pure water by intragastric gavage daily. The mice in both the normal-fat diet and high-fat diet groups were also gavaged with 200 µL of sterilized water daily. A group of two caged mice was used to exclude the co-housing effect of the gut microbiota. At the end of the study, mice that were fasted overnight were anesthetized and killed. Blood samples were collected, and serum samples were separated by centrifuging the blood at 3000 rpm and 4 °C for 10 min. The entire liver and ileum were dissected and weighed, some sections were snap-frozen in liquid nitrogen before storage at −80 °C, and some sections were fixed in paraformaldehyde for further histological analysis. All animal procedures in this study were approved by the Animal Care and Use Committee of Southern Medical University.

### 2.4. Fecal Microbiome Transplantation

FMT was performed based on an established protocol [16]. Briefly, 6-week-old male donor mice were randomly fed a HFD with or without 150 mg/kg PLE. Stools were collected daily from donor mice and pooled. Donor stools (50 mg) were diluted with saline, homogenized for 1 min using a vortex to achieve a liquid slurry, and then centrifuged at 2000× *g* and 4 °C for 1 min to remove particulates. The supernatant was centrifuged at 12,000× *g* and 4 °C for 5 min and discarded to obtain the precipitate. Next, 600 μL of saline was added to resuspend the precipitate, and 200 μL of bacterial resuspension liquid was transplanted into recipient mice fed a HFD for 8 weeks. A group of two caged mice was used to exclude the cocage effect of the gut microbiota. The outcomes of the mice were blindly collected.

### 2.5. Real-Time PCR

The total RNA was extracted from 100 mg of liver tissue using TRIzol™ (CURATE Biotechnology, Carlsbad, CA, USA). In this case, cDNA was synthesized using a reverse transcription kit (ACCURATE BIOLOGY, Changsha, China), and gene expression was quantified using a SYBR Green Polymerase Chain Reaction (PCR) kit with an Applied Biosystems ViiA 7 Dx instrument. A reverse transcription PCR was performed to quantify the expression of Toll-like receptor 4 (TLR4), cluster of differentiation 14 (CD14), myeloid differentiation factor 88 (MyD88), tumor necrosis factor-α (TNFα), interleukin 10 (IL-10) and inducible nitric oxide synthase (iNOS) in the livers of mice in each group. All the primers were from BGI, and the primer sequences are shown in Table 1. The internal control used was glyceraldehyde-3-phosphate dehydrogenase (GAPDH) mRNA.

### 2.6. Western Blotting

For total protein extraction, RIPA lysis buffer (KGP2100, Keygen Biotech, Shanghai, China) with a protease inhibitor cocktail (KGP2100, Keygen Biotech, Nanjing, China) and a phosphatase inhibitor mixture (KGP2100, Keygen Biotech, Nanjing, China) were utilized to lyse the tissues. The protein concentrations were detected using the BCA Protein Assay Kit (KGP902, Keygen Biotech, Nanjing, China), and Western blotting was performed using sodium dodecyl sulfate‒polyacrylamide gel electrophoresis (SDS‒PAGE). The transferred membranes were incubated at 4 °C overnight with the following primary antibodies: mouse anti-*TLR4* antibody (1:1000, 16c5074; Abcam, Cambridge, UK), rabbit anti-*MyD88* (1:1000, ab133739; Abcam, China) and rabbit anti-*β actin* (1:1000, ab8226; Abcam, Cambridge, UK). The membranes were then incubated with the corresponding secondary antibodies, including HRP antirabbit antibody (1:10,000, SA00001-1/SA00001-2, Proteintech, Chicago, IL, USA), for 1 h at room temperature. The blots were visualized using ECL chemiluminescent liquid (KGP1127, Abcam, Cambridge, UK) and an ultrasensitive chemiluminescence instrument (Bio-Rad, Hercules, CA, USA), and the densities were analyzed using Image-Pro Plus 6.0 (Media Cybernetics, Rockville, MD, USA).

### 2.7. Histological Staining

After the experiment, the liver and ileum tissue were dissected and fixed with 4% paraformaldehyde solution. The fixed tissues were dehydrated, embedded in paraffin, and sectioned to a thickness of 4 μm. The sections were stained with hematoxylin and eosin and examined using an optical microscope (Nikon Eclipse ci, Nikon digital sight DS-FI2, MADE IN JAPAN). The statistical evaluation of the nonalcoholic fatty liver disease (NAFLD) activity scores (NASs) was blindly performed by independent specialists [17].

### 2.8. Serum Inflammation Assay

The serum was obtained by centrifugation (4000 rpm, 4 °C, 10 min). The lipopolysaccharide (LPS), TNF-α and interleukin-6 (IL-6) levels in the serum of mice were measured using ELISA kits (CUSABIO, Houston, TX, USA) following the manufacturer’s instructions.

### 2.9. DNA Extraction and 16S rRNA Gene Sequencing

The feces of the mice in each group were collected before intervention and after the experiment and frozen at −80 °C after collection. DNA was extracted from the feces using a QIAamp DNA Mini Kit (QIAGEN GmbH, Hilden, Germany), and the V4 region of the bacterial 16S rRNA was amplified by PCR (95 °C for 2 min followed by 25 cycles at 95 °C for 30 s, 55 °C for 30 s, and 72 °C for 30 s and a final extension at 72 °C for 5 min) using the primers 5′-ACTCCTACGGGAGGCAGCA-3′ for 338F F and 5′-GGACTACHVGGGTWTCTAAT-3′ for 806R. All PCR amplicons were mixed and sequenced using the Illumina NovaSeq 6000/MiSeq according to the manufacturer’s protocol.

### 2.10. Bioinformatics Processing

The raw sequencing data were preprocessed using a pipeline (https://github.com/SMUJYYXB/GGMP-Regional-variations, accessed on 1 May 2021) as previously described, and Dada2 was performed to generate amplicon sequence variants (ASVs) [18]. PyNAST and FastTree were used to align the sequences and build a phylogenetic tree [19,20]. The RDP classifier in QIIME (version 1.9.1) with the Greengenes (version 13.8) database was used for taxonomic assignment [21,22,23]. Based on the ASVs, we calculated the alpha-diversity values (Phylogenetic Diversity (PD) Whole Tree Index, Chao1 Index and observed ASVs). Beta-diversity measurements and principal coordinate analyses (PCoAs) based on unweighted UniFrac distance metrics were performed with QIIME (version 1.9.1). The differences in the beta diversity between groups were detected by permutational multivariate analysis of variance (PERMANOVA) [24,25]. For linear discriminant analysis effect size (LEfSe) [26], the biological relevance and statistical significance were considered, and identification was performed to differentially represent the level of classification among groups. The niche-specific relational networks and specific “bacterium-metabolite” relational networks were visualized using Cytoscape software version 3.9.1 (The Cytoscape ConsortiumSan, San Diego, CA, USA; https://cytoscape.org/, accessed on 1 January 2022), and Spearman correlation analyses of both networks were performed. Niche-specific relational networks mainly reflect the relationship between the genus and serum or fecal metabolite biomarkers. The genus with LEfSe LDA > 2 among groups was selected, and the correlation coefficients among them were calculated to reflect the correlation between species at the genus level. The colors of the lines indicate positive and negative correlations; red indicates a positive correlation, and blue indicates a negative correlation. The specific “bacterium-metabolite” relational networks show all the significant “bacteria genus-serum/fecal metabolite” pairs selected based on FDR ≤ 0.05.

### 2.11. Metabolomics Profiling of Serum and Fecal Samples

Metabolomics analyses of serum and fecal samples were performed as described previously with minor modifications [27,28]. Briefly, a 50-μL serum sample was precipitated with 150 µL of ice-cold acetonitrile, and the mixed solution was then centrifuged at 15,000× *g* and 4 °C for 15 min. The supernatant was transferred to a new clean tube and dried with nitrogen. The residue was reconstituted in 50 μL of water:acetonitrile (1:1) solution and centrifuged at 15,000× *g* and 4 °C for 15 min. Subsequently, 2 μL of supernatant was subjected to UPLC/Q-TOF MS analysis. In order to ensure data quality for metabolic profiling, pooled quality control samples were prepared by mixing equal (5 µL) amounts of each serum sample. Similarly, for fecal samples, 50 mg of lyophilized feces was extracted with 500 µL of ice-cold ultrapure water-acetonitrile (1:1). After centrifugation at 15,000× *g* and 4 °C for 15 min, the supernatants were combined and dried with nitrogen. The residue was further dissolved in 100 µL of water:acetonitrile (1:1) solution and centrifuged at 15,000× *g* and 4 °C for 15 min. Next, 1 μL of supernatant was subjected to UPLC/Q-TOF MS analysis. The serum and feces metabolome were acquired using a Waters ACQUITY™ UHPLC system coupled with a SYNAPT G2-Si high-definition Q-TOF mass spectrometer (Waters, Manchester, UK). Chromatographic separations were performed on an ACQUITY™ HSS T3 C18 column (100 mm × 2.1 mm i.d., 1.8 μm) at 40 °C. The mobile phases comprised acetonitrile modified with 0.1% formic acid (A) and 0.1% aqueous formic acid (B) at a flow rate of 0.3 mL/min. The gradient program was as follows: isocratic 3% A over 0–1 min, linear 3%−70% A over 1–8 min, 70% A over 8–10 min, 70%–90% A over 10–17 min, 90%−100% A over 17–18 min, and isocratic 100% A over 18–21 min. The LC eluent was introduced to a SYNAPT G2-Si MS equipped with an electrospray ionization source (ESI). Q-TOF MS was operated in the negative and positive ion modes, and the key parameters were set as follows: capillary voltage, 3.0/−2.5 kV, sample cone voltage, 40 V, source temperature, 120 °C, desolvation temperature, 500 °C, nitrogen gas flow, 900 L/h, cone gas flow, 10 L/h, and TOF acquisition rate, 0.3 s/scan. The MS data were collected in the centroid mode with a mass range from *m*/*z* 100 to *m*/*z* 1200 using the MSE scan mode.

### 2.12. Statistical Analysis

The results of the biological assay are presented as the means ± SEs. The differences between two or more groups were analyzed by the Wilcoxon rank sum test and corrected using the method developed by Benjamini and Hochberg. The statistical analysis processing and data visualization were performed in R using the ggplot2 and vegan packages. Spearman correlations were calculated by linear regression to define pairwise associations between study variables. *p* ≤ 0.05 was considered to indicate statistical significance in all the analyses. The acquired raw UPLC/Q-TOF MS data of serum and fecal samples were processed using MS DIAL software [29]. The differentiating features with the greatest contribution to the separation were extracted according to their variable importance in the projection (VIP) values (>1.0, *p* < 0.05). These metabolite peaks were tentatively assigned by the MS and MS/MS data of endogenous mammalian metabolites acquired from available databases, such as the Human Metabolome Database (HMDB, http://www.hmdb.ca, accessed on 1 May 2021), METLIN (http://metlin.scripps.edu, accessed on 1 May 2021).

## 3. Results

### 3.1. PLE Prevents HFD-Induced MAFLD Associated with the Gut Microbiota

In this study, the beneficial effects of PLE on HFD-induced MAFLD in ApoE^−/−^ mice receiving PLE supplementation with a HFD or NFD were investigated. The livers of the mice in each group were observed with the naked eye, and the observations showed that the livers of the mice in the HFD group were yellower than those of the NFD group, whereas the livers of the HFD+PLE mice had a healthier red and compact appearance compared with those of the HFD group (Appendix A). Moreover, PLE administration significantly reduced liver fat vacuoles and inflammatory infiltration in mice receiving a HFD in line with a lower MAFLD activity score (Figure 1a), which indicated the preventative effects of PLE on HFD-induced MAFLD.

An analysis of the chemical composition of PLE by UPLC‒MS/MS (Appendix A and Table 2) identified 33 phenolic compounds, including flavonoids, catechin and epicatechin derivatives, based on MS and MS/MS spectral information, and many polyphenols have been demonstrated to alter the composition of gut microbiota [30,31]. Therefore, in this study, we investigated whether the effects of PLE on HFD-induced MAFLD were associated with the gut microbiota. The HFD significantly reduced the richness of the gut microbiota compared with that of the NFD group, and this effect was reversed by PLE supplementation in HFD-fed mice, as evidenced by the α-diversity analysis (Figure 1b). In addition, before the experiment, no significant difference in the distance on the PCoA map was found among all the groups (Figure 1c). After 3 months of PLE intervention, a PCoA showed that the microbial signatures among the groups were different (Figure 1d). The Lefse analysis showed lower abundances of *Rikenellaceae, Odoribacter, RF39, Alistipes, Mogibacteriaceae, Faecalibacterium* and *Turicibacter* and higher abundances of *Roseburia, Bilophila*, *Helicobacter* and *Lachnospiraceae* in mice receiving the HFD compared with the mice in the NFD and HFD+PLE groups (Figure 1e,f).

Furthermore, a correlation analysis of abundances of genera in the gut microbiota with the MAFLD activity score revealed that the *Rikenellaceae, Odoribacter, RF39, Alistipes, Mogibacteriaceae, Faecalibacterium* and *Turicibacter* abundances were negatively correlated MAFLD and that *Roseburia, Bilophila, Helicobacter* and *Lachnospiraceae* were positively correlated MAFLD (Figure 1g). The abovementioned results suggest that PLE could reshape the microbiota potential of HFD-fed mice.

### 3.2. PLE Prevents HFD-Induced MAFLD Associated with Microbial-Derived Metabolites 

Microbiota-derived metabolites have been strongly implicated in the pathogenesis of host metabolic health [32]. In order to investigate the metabolites responding to the PLE-altered gut microbiota underlying the ameliorated MAFLD, we next performed an untargeted metabolomics analysis of serum and fecal samples.

Orthogonal partial least squares-discrimination analysis (OPLS-DA) showed the different metabolomic signatures in either serum or feces among groups (Appendix A). Notably, phosphatidyl inositol (PI), phosphatidylserine (PS), phosphatidylethanolamine (PE) and phosphatidylcholine (PC) were enriched in the HFD group compared with the NFD or HFD+PLE group (Figure 2a). Low-inflammatory-related metabolites, such as lysophospholipids (LysoPCs) [33], were enriched in the HFD group, but were significantly reduced in the NFD and HFD+PLE groups (Figure 2a). In addition, fecal metabolomics showed that 3-methoxybenzenepropanoic acid, O-ureidohomoserine, 7-sulfocholic acid and chenodeoxycholic acid sulfate were significantly enriched in the feces of HFD-fed mice, whereas N-lauroyl cysteine, taurine, prolyl-tryptophan, 5-phenylvaleric acid, 2-dodecylbenzenesulfonic acid and cholesterol sulfate were significantly increased in the HFD+PLE group (Figure 2b).

The study further identified which bacteria were associated with these key differential metabolites in serum/feces through a Spearman correlation analysis. The genera *Odoribacter, Alistipes, Christensenellaceae, Rikenellaceae, R39* and *Mogibacteriaceae* were significantly negatively correlated with serum LysoPCs, and these genera were enriched in the NFD and HFD+PLE groups (Figure 2c). The genera *Roseburia* and *Lachnospiraceae*, which were significantly positively correlated with serum LysoPCs, were enriched in the HFD group (Figure 2c). Furthermore, *Rikenellaceae* was significantly positively correlated with fecal taurine, which has the potential to protect the intestinal barrier, whereas *Lachnospiraceae* was significantly negatively correlated with fecal taurine (Figure 2d). These results show that PLE-modulated metabolites are associated with the PLE-altered gut microbiota.

### 3.3. PLE Alleviates Intestinal Barrier Permeability and Inhibits LPS-Related TLR4-MyD88 Signaling in the Liver

HFD could also affect the epithelial integrity and thus lead to impaired gut permeability and the release of lipopolysaccharide into the circulation. Therefore, we subsequently investigated the effects of PLE on the intestinal barrier and found that PLE improved the ileal morphology in mice receiving a HFD, as evidenced by thickening of the intestinal wall and increased villi (Figure 3a). Similarly, PLE significantly increased the gene expression of Claudin-4 and Occludin, which are essential genes in the tight junctions of the intestine, in the ileum of mice receiving a HFD (Figure 3b). Moreover, PLE restored HFD-induced systemic inflammation by reducing the serum LPS, TNF-α, and IL-6 levels (Figure 3c) but also reversed LPS-induced downstream inflammatory signaling of TLR4 and MyD88 at both the mRNA (Figure 3d) and protein levels (Figure 3e) in the livers of mice receiving an HFD. Together, our results indicate that PLE alleviates intestinal barrier permeability and inhibits LPS-related TLR4-MyD88 signaling in the liver.

### 3.4. FMT from PLE-Treated Donor HFD-Fed Mice Prevents HFD-Induced MAFLD in Recipient Mice

In order to further unravel the causal role of the gut microbiota in the beneficial effect of PLE on MAFLD, we subsequently performed a FMT study (FMT, Figure 4a). As expected, a PCoA based on unweighted UniFrac distance matrices showed a significant effect between the intestinal microbiota of donors and recipient mice (Figure 4b). From the donors, 68 of 103 genera successfully colonized the intestine of the recipient mice (Figure 4c). Additionally, 60.29% of the shared genera were found to show the same varying trend across the microbiota of donors and recipient mice (Figure 4d). These data collectively confirmed successful FMT. In addition to finding the different compositions of the gut microbiota in different groups of recipient mice (Figure 4e,f), we observed ameliorated MAFLD (Figure 4g and Appendix A) in the livers of the mice receiving FMT from PLE-treated donor mice, which indicated that the beneficial effects of PLE on HFD-induced MAFLD are causally mediated by the gut microbiota.

Notably, we found that the abundances of the bacteria *Rikenellaceae*, *Odoribacter*, and *Helicobacter* (Figure 4f), which were significantly regulated by PLE administration (Appendix A), was also markedly changed in the recipient mice receiving FMT from PLE-treated donor mice, which suggested that these genera may be the core bacteria in PLE-alleviated MAFLD.

### 3.5. FMT from PLE-Treated Donor Mice Alters Metabolites, Alleviates Intestinal Barrier Permeability and Inhibits Liver Inflammation in Recipient Mice with HFD-Induced MAFLD

In order to reveal whether the PLE-regulated metabolites are also altered by the associated FMT, we analyzed the metabolomics in the serum (Figure 5a) and fecal samples (Figure 5b) of recipient mice receiving FMT. The gut microbiota indeed mediates the effects of PLE on regulating metabolites, as evidenced by separate dot plots according to PLS-DA (Appendix A).

In serum, LysoPCs, taurocholic acid, tauroursodeoxycholic acid, and 3−sulfodeoxycholic acid were significantly enriched in the HFD group compared with the NFD and HFD+PLE groups (Figure 5a). In the feces, the 3-sulfodeoxycholic acid and 12−Ketodeoxycholic acid levels were significantly higher in the HFD-FMT group than in the HFD+PLE-FMT group (Figure 5b).

In addition, PLE-derived FMT alleviated intestinal barrier permeability (Figure 5c) by significantly increasing the gene expression of ZO-1 and occludin, but not claudin-4 in the ileum (Figure 5d) and reduced liver inflammation by inhibiting serum inflammatory cytokines, i.e., LPS, TNFα, and IL-6 (Figure 5e), and TLR4-MyD88 signaling in the liver (Figure 5f,g). Altogether, these findings suggest that the gut microbiota mediates the gut barrier protection and anti-inflammatory properties of PLE.

## 4. Discussion

The study demonstrated that PLE effectively prevented HFD-induced hepatic steatosis in ApoE^−/−^ mice and suggested that PLE treatment could protect against MAFLD by improving the intestinal barrier function and reducing the LPS levels and liver TLR4/MyD88-mediated inflammation in HFD-induced ApoE^−/−^ mice. Horizontal-FMT experiments demonstrated that the gut microbiota was a key target of PLE to improve MAFLD. Further research showed that FMT from PLE-and-HFD-treated donor mice could significantly reduce the sulfation of secondary bile acids such as 3-sulfodeoxycholic acid compared with FMT from HFD-treated donor mice. To the best of our knowledge, this study provides the first clarification that PLE can prevent MAFLD by modulating the gut microbiota.

The study showed that the LPS/TLR4 signaling pathway played an important role in the mechanism through which PLE prevents MAFLD. Small intestinal bacterial overgrowth is always observed in MAFLD patients and may increase the permeability of the intestine, leading to translocation of bacteria and their byproducts, such as LPS [34]. After reaching the liver, LPS is ingested by hepatocytes and Kupffer cells and then upregulates TLR4 signaling [35]. Stimulation of the LPS-TLR4 signaling pathway induces the interaction of TLR4 with the adaptor molecule MyD88 and eventually leads to the release of proinflammatory mediators such as TNF-α [36], which ultimately causes inflammation and progression of MAFLD to NASH and subsequent fibrosis [37,38]. In this study, PLE reduced gut dysbiosis in HFD-fed mice accompanied by reduced serum LPS, TNFα and IL-6 levels and downregulation of the hepatic TLR4-MyD88 signaling pathway. Moreover, the serum metabolomics data revealed that the level of the chronic inflammation biomarker LysoPCs in the serum of mice in the HFD+PLE group was reduced, which also confirmed that PLE and the microflora regulated by PLE exerted anti-inflammatory effects on MAFLD.

FMT has great therapeutic potential for inflammatory bowel disease, chronic gastrointestinal infections, and chronic liver disease [39]. Our study found that recipient mice receiving FMT from HFD+PLE mice also showed a remission of MAFLD, which supports the beneficial effects of PLE that can be partially mediated by gut microbiota. In addition, our study found that PLE could increase the alpha diversity of then gut microbiota in HFD-fed mice, which always presents more pronounced dysmetabolism and low-grade inflammation [40]. Furthermore, a correlation analysis of abundances of genera in the gut microbiota with the MAFLD activity score revealed that the abundances of *Rikenellaceae* and *Odoribacter*, which were enriched in the HFD+PLE and HFD+PLE-FMT groups, were negatively correlated with MAFLD and that *Helicobacter*, which was enriched in the HFD and HFD-FMT groups, was positively correlated with MAFLD. This result is in line with the clinical findings that *Rikenellaceae* is decreased in MAFLD patients [2]. In addition, a lower relative abundance of *Odoribacter* was found to be associated with MAFLD [41]. Members of the *Odoribacteraceae* and *Rikenellaceae* families primarily express the butyrate kinase gene, which is the pathway used in most intestinal ecosystems to produce butyrate [42]. Therefore, the decreased abundance of the *Odoribacter* genus and *Rikenellaceae* family may result in reduced butyric acid availability, leading to host inflammation [42,43].

Similar to our results, a recent murine study involving metagenomics reported that in mice exposed to an HFD, dysbiosis is characterized by an increase in the Firmicutes/Bacteroidetes ratio and gram-negative bacteria and significantly increased detection of the *Helicobacter* genus [44]. The presence of *Helicobacter* in the intestine has been suggested to improve gut permeability and promote bacterial LPS through the portal vein to the liver, and these effects are accompanied by increased levels of LPS-mediated inflammatory cytokines in the liver, which enhance hepatic inflammation and fibrosis [45,46]. These reports collectively suggest that the *Helicobacter* genus is likely to be the core genus in the mechanism through which PLE ameliorates MAFLD.

LPS concentrations are highest in the gut lumen, where many trillions of commensal bacteria reside. A defective TJ barrier allows paracellular flux of LPS when intestinal permeability disorders occur [47,48]. Both PLE and the intestinal flora from HFD-fed mice regulated by PLE exert a protective effect on the intestinal barrier of HFD-fed mice, but which microbiota metabolites are regulated by these treatments to protect the intestinal barrier are unclear. PLE administration enriched metabolites that protect the intestinal barrier, including N-lauroyl cysteine, prolyl−tryptophan, and taurine, in the intestine of HFD-fed mice. N-Lauroyl cysteine could reduce the content of oxidative stress factors such as glutathione and malondialdehyde in the intestinal wall tissue of rats after partial hepatectomy [49]. Prolyl−tryptophan is one of the microbial tryptophan metabolites that reportedly regulate gut barrier function via the aryl hydrocarbon receptor [50]. Taurine also reportedly regulates mucosal barrier function to alleviate LPS-induced duodenal inflammation in chickens and mice [51,52]. Moreover, the Spearman correlation analysis first revealed that the genus *Rikenellaceae* was significantly positively correlated with fecal taurine, and the connection between *Rikenellaceae* and taurine was further explored. Although we did not find a similarity in the changes in fecal metabolites upon PLE treatment and PLE-derived FMT, we found that mice receiving PLE-derived FMT had a lower level of secondary bile acids, i.e., 3-sulfodeoxycholic acid, in both serum and feces. It is known that secondary bile acids exert anti-inflammatory effects [53], and the effects are less visible when the secondary bile acids are sulfated [54]. Therefore, the decreased levels of sulfated bile acids in the gut might explain how PLE-associated gut microbiota ameliorates inflammation, but this finding remains to be further verified. The results from the nontarget metabolome analysis of feces suggested that PLE and the bacterial flora from the mice in the HFD+PLE group mediate different microbial metabolites to improve the intestinal barrier in HFD-fed mice. A possible explanation is that PLE is a mixture, and many microbial metabolites regulated by PLE may play an important role in gut barrier protection.

According to a study conducted by Gong et al. [55], PLE has higher epigallocatechin and epicatechin contents than extracts of other dark teas, but whether this finding is key to the PLE-mediated regulation of the microbiota and improvement of MAFLD remains to be further explored. In addition, chronic exposure to a HFD increases oxygenation in the colon by altering the intestinal epithelial physiology, resulting in an increased abundance of *Escherichia coli* and leading to gut dysbiosis [56]. Considering the limited bioavailability of tea polyphenols, a significant fraction of tea polyphenols can enter the colon, where they are exposed to the gut microbial community [57]. Moreover, tea polyphenols show a strong antioxidant capacity due to their multiple phenolic hydroxyl groups and the main structure of 2-phenyl-benzopyran [58,59]. Therefore, we speculate that PLE could balance the oxidative stress in the colonic lumen of HFD-fed mice, which may be the reason why PLE could improve gut dysbiosis in HFD-fed mice. Since the gut microbiota plays a key role in the improvement of NAFLD induced by PLE, further exploration is needed in order to determine whether or not PLE exerts a regulatory effect on the gut microbiota of NAFLD patients. Second, a metagenomics investigation is needed to determine which core bacterial strains are regulated by PLE. Moreover, the mechanism through which bacterial strains protect the intestinal barrier and alleviate NAFLD remains to be further explored.

## 5. Conclusions

This study lays the foundation for understanding the complex crosstalk among polyphenol-rich Liupao tea extract (PLE), the gut microbiota and the host. In summary, these findings collectively indicate the therapeutic potential of PLE in combating MAFLD as a novel prebiotic.

## Figures and Tables

**Figure 1 nutrients-14-04930-f001:**
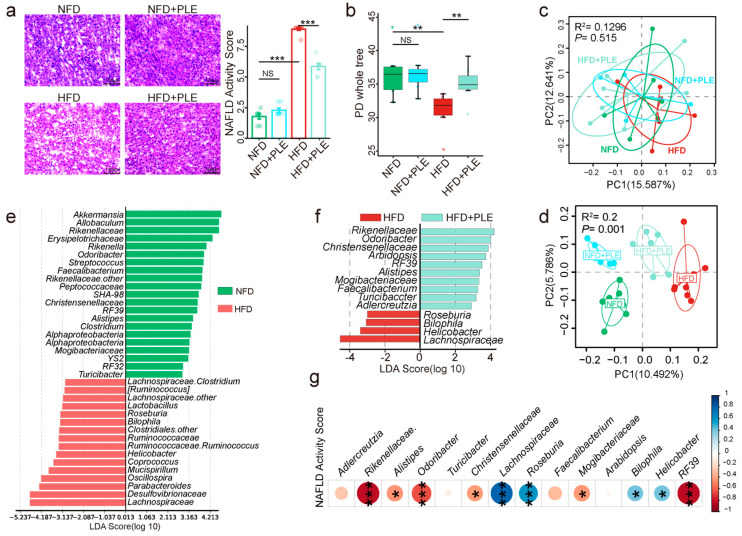
PLE prevents HFD-induced NAFLD associated with the gut microbiota. (**a**) Representative H&E images of the liver and NAS scores in each group (n = 7~9). (**b**) Comparison of alpha-diversity indices (PD whole tree index) in each group (n = 7~9). (**c**) Principal coordinate analysis (PCoA) plots of each group before treatment based on unweighted UniFrac distance matrices (n = 7~9). (**d**) PCoA plots of each group after treatment based on unweighted UniFrac distance matrices (n = 7~9). (**e**) Linear discriminant analysis effect size (LEfSe) analysis between the NFD and HFD groups (n = 7~9). (**f**) LEfSe analysis between the HFD and HFD+PLE groups (n = 7~9). (**g**) Correlation analysis of the abundances of genera in the gut microbiota with the NAFLD activity score. *p* values were calculated using the Wilcoxon rank-sum test and corrected by the method described by Benjamini and Hochberg, * *p* < 0.05, ** *p* < 0.01 and *** *p* < 0.01 indicate a significant difference between the two corresponding groups. NFD: treatment with a normal-fat diet; HFD: treatment with a high-fat diet; NFD+PLE: treatment with a normal-fat diet and 150 mg/kg polyphenol-rich Liupao tea extract; HFD+PLE: treatment with a high-fat diet and 150 mg/kg polyphenol-rich Liupao tea extract.

**Figure 2 nutrients-14-04930-f002:**
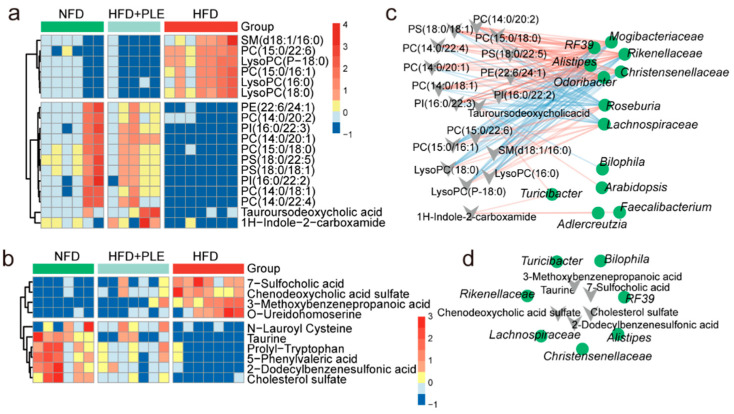
PLE prevents HFD-induced NAFLD associated with metabolites. (**a**) Heatmap visualizing the changes in the contents of overlapping biomarkers in the serum of the NFD, HFD and HFD+PLE groups (n = 5~7). (**b**) Heatmap visualizing the changes in the contents of overlapping biomarkers in the feces of the NFD, HFD and HFD+PLE groups (n = 6~7). (**c**) Spearman correlation analysis between the 14 genera found by Lefse between the HFD and HFD+PLE groups and serum metabolite biomarkers in mice in the NFD, HFD and HFD+PLE groups. (**d**) Spearman correlation analysis between the 14 genera found by Lefse between the HFD and HFD+PLE groups and fecal metabolite biomarkers in mice in the NFD, HFD and HFD+PLE groups. *p* values were calculated using the Wilcoxon rank-sum test and corrected by the method described by Benjamini and Hochberg. NFD: treatment with a normal-fat diet; HFD: treatment with a high-fat diet; NFD+PLE: treatment with a normal-fat diet and 150 mg/kg polyphenol-rich Liupao tea extract; HFD+PLE: treatment with a high-fat diet and 150 mg/kg polyphenol-rich Liupao tea extract.

**Figure 3 nutrients-14-04930-f003:**
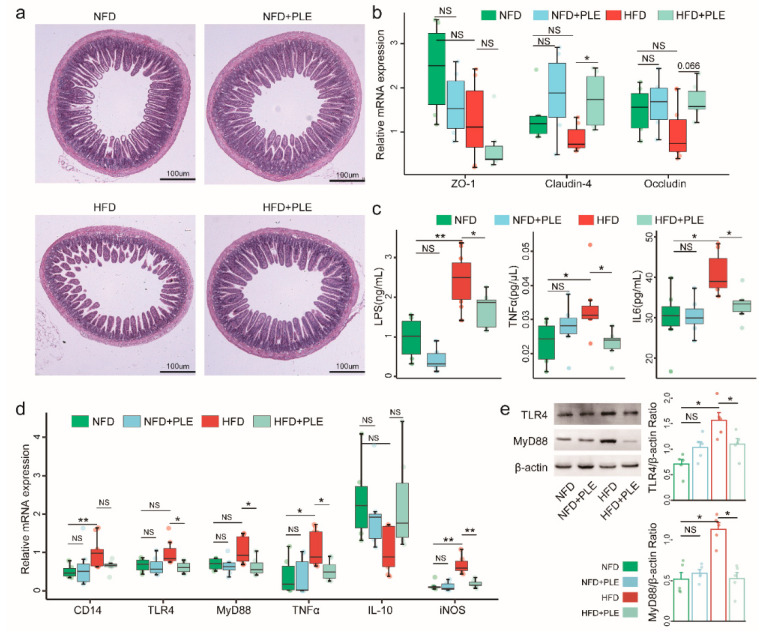
PLE alleviates intestinal barrier permeability and inhibits LPS-related TLR4-MyD88 signaling in the liver. (**a**) Representative H&E images of the terminal ileum in each group of mice (n = 7~9). (**b**) mRNA expression of ZO-1, claudin-4 and occludin in the ileum of each group of mice (n = 7~9). (**c**) Levels of serum LPS, TNFα and IL-6 in each group of mice (n = 7~9). (**d**) mRNA levels of the TLR4-MyD88 signaling pathway in the liver of each group of mice (n = 7~9). (**e**) Protein levels of TLR4 and MyD88 in the liver of each group of mice (n = 5). *p* values were calculated using the Wilcoxon rank-sum test and corrected using the method described by Benjamini & Hochberg, * *p* < 0.05 and ** *p* < 0.01 indicate a significant difference between the two corresponding groups. NFD: treatment with a normal-fat diet; HFD: treatment with a high-fat diet; NFD+PLE: treatment with a normal-fat diet and 150 mg/kg polyphenol-rich Liupao tea extract; HFD+PLE: treatment with a high-fat diet and 150 mg/polyphenol-rich Liupao tea extract.

**Figure 4 nutrients-14-04930-f004:**
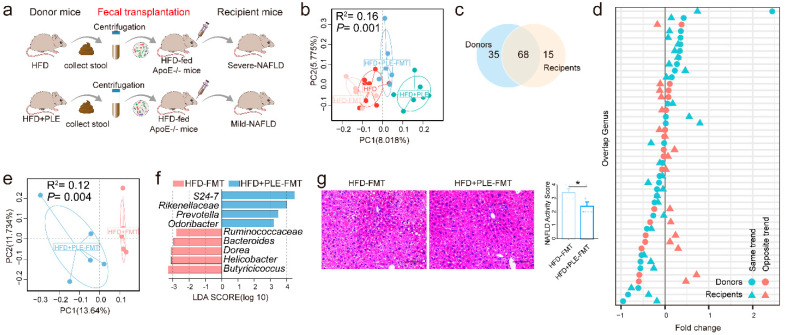
FMT from PLE-treated donor mice prevented HFD-induced NAFLD in recipient mice. (**a**) Study design of the fecal transplantation experiment. (**b**) Principal coordinate analysis (PCoA) plots of donors and recipient mice based on unweighted UniFrac distance matrices (n = 6). (**c**) Venn diagram comparing the numbers of shared genera in the gut microbiome of donors and recipient mice (n = 6). (**d**) Concordance of genus variations between the intestinal microbiota of donors and recipient mice. (**e**) PCoA plots of the HFD-FMT and HFD+PLE-FMT groups based on unweighted UniFrac distance matrices. (**f**) Linear discriminant analysis effect size (LEfSe) analysis identified different taxa between the HFD-FMT and HFD+PLE-FMT groups (n = 6). The LDA scores (log10) > 2.0 are listed. (**g**) Representative H&E images of liver and NAFLD activity scores of the HFD-FMT and HFD+PLE-FMT groups (n = 7). *p* values were calculated using the Wilcoxon rank-sum test and corrected using the method described by Benjamini and Hochberg, * *p* < 0.05 indicate a significant difference between the two corresponding groups. HFD-FMT: fecal microbiota transplantation from mice in the HFD group and treatment with a high-fat diet; HFD+PLE-FMT: fecal microbiota transplantation from mice in the HFD+PLE group and treatment with a high-fat diet.

**Figure 5 nutrients-14-04930-f005:**
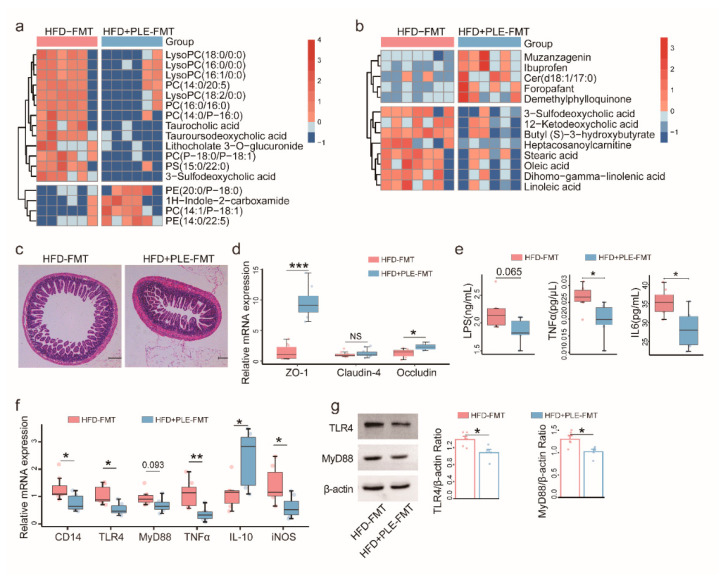
FMT from PLE-treated donor mice alters metabolites, alleviates intestinal barrier permeability and inhibits liver inflammation in recipient mice with HFD-induced MAFLD. (**a**) Heatmap visualizing the changes in the contents of overlapping biomarkers in the serum of the HFD-FMT and HFD+PLE-FMT groups (n = 6). (**b**) Heatmap visualizing the changes in the contents of overlapping biomarkers in the feces of the HFD-FMT and HFD+PLE-FMT groups (n = 7). (**c**) Representative H&E images of the terminal ileum of the HFD-FMT and HFD+PLE-FMT groups (n = 7). (**d**) mRNA expression of zonula occludens-1, claudin-4 and occludin in the ileum of the HFD-FMT and HFD+PLE-FMT groups (n = 7). (**e**) Levels of serum LPS, TNFα and IL-6 in the HFD-FMT and HFD+PLE-FMT groups (n = 7). (**f**) mRNA levels of the TLR4-MyD88 signaling pathway in the livers of the HFD-FMT and HFD+PLE-FMT groups (n = 7). (**g**) Protein levels of TLR4 and MyD88 in the livers of the HFD-FMT and HFD+PLE-FMT groups (n = 5). *p* values were calculated using the Wilcoxon rank-sum test and corrected using the method described by Benjamini and Hochberg, * *p* < 0.05, ** *p* < 0.01 and *** *p* < 0.01 indicate a significant difference between the two groups. HFD-FMT: fecal microbiota transplantation from mice in the HFD group and treatment with a high-fat diet; HFD+PLE-FMT: fecal microbiota transplantation from mice in the HFD+PLE group and treatment with a high-fat diet.

**Table 1 nutrients-14-04930-t001:** RT-qPCR primer sequences.

Primer Name Sequence	Sequence
iNOS-F	5′-GGAGTGACGGCAAACATGACT-3′
iNOS-R	5′-TCGATGCACAACTGGGTGAAC-3′
TNF-α-F	5′-CCTGTAGCCCACGTCGTAG-3′
TNF-α-R	5′-GGGAGTAGACAAGGTACAACCC-3′
IL-10-F	5′-AGCCTTATCGGAAATGATCCAGT-3′
IL-10-R	5′-GGCCTTGTAGACACCTTGGT-3′
Myd88-F	5′-GTTGTGTGTGTCCGACCGT-3′
Myd88-R	5′-GTCAGAAACAACCACCACCATGC-3′
TLR4-F	5′-CCTCTGCCTTCACTACAGAGACTTT-3′
TLR4-R	5′-TGTGGAAGCCTTCCTGGATG-3′
CD14-F	5′-GGAAGCCAGAGAACACCATC-3′
CD14-1-R	5′-CCAGAAGCAACAGCAACAAG-3′
GAPDH-F	5′-CTGCGACTTCAACAGCAACT-3′
GAPDH-R	5′-GAGTTGGGATAGGGCCTCTC-3′
Ocludin-F	5′-ACGGACCCTGACCACTATGA-3′
Ocludin-R	5′-TCAGCAGCAGCCATGTACTC-3′
Claudin-4-F	5′-ATCGTTGTCCGCGAGTTCTA-3′
Claudin-4-R	5′-GCTTGTCGTTGCTACGAGGT-3′
ZO-1-F	5′-GCTTTAGCGAACAGAAGGAGC-3′
ZO-1-R	5′-TTCATTTTTCCGAGACTTCACCA-3′

**Table 2 nutrients-14-04930-t002:** Tentative LC‒MS identification of the small molecules from polyphenol-rich Liupao extract (PLE).

NO	Rt (min)	Precursor Ion (*m/z*)	MS/MS (*m/z*)	Molecular Formula	Tentative Identification
1	0.86	191.0631	191.0631, 149.0078	C_10_H_8_O_4_	Noreugenin
2	0.93	205.0509	205.0509, 107.0124	C_11_H_10_O_4_	Eugenin
3	1.27	175.1241	175.1241, 1130202, 87.0072	C_6_H_8_O_6_	Ascorbic acid
4	1.8	169.0218	169.0218, 125.0303	C_7_H_6_O_5_	Gallic acid
5	1.89	343.0541	343.0541, 191.0525, 169.0137	C_14_H_16_O_10_	5-Galloylquinic acid
6	3.06	305.0686	305.0686, 137.0245, 125.0244	C_15_H_14_O_7_	Epigallocatechin
7	3.36	353.0869	353.0869, 309.0918, 179.0295	C_16_H_18_O_9_	Chlorogenic acid
8	3.58	451.123	451.1230, 305.0630, 125.0239	C_24_H_20_O_9_	Epigallocatechin 3-O-p-coumarate
9	3.71	577.1368	577.1368, 289.0727	C_30_H_26_O_12_	Procyanidin B2
10	3.78	337.1025	275.0536, 163.0385	C_16_H_18_O_8_	Coumaroylquinic acid
11	3.86	289.1124	289.1124, 245.1068, 123.0456	C_15_H_14_O_6_	Catechin
12	3.88	287.0935	287.0935, 269.0812, 151.1012	C_15_H_12_O_6_	Dihydrokaempferol
13	4.03	353.0878	353.0878, 191.0331, 175.0031, 147.0419	C_16_H_18_O_9_	Scopolin
14	4.05	577.1346	577.1346, 425.0778, 289.0706, 151.0387	C_30_H_26_O_12_	Procyanidin B4
15	4.05	425.085	425.0850, 407.0778, 169.0137	C_22_H_18_O_9_	Epiafzelechin 3-gallate
16	4.05	431.0956	431.0956, 413.0873, 269.0456	C_21_H_20_O_10_	Anthemoside
17	4.09	205.051	205.0510, 189.0131, 175.0490	C_11_H_10_O_4_	Eugenin
18	4.35	289.1023	289.1023, 245.0880, 125.0249, 109.0295	C_15_H_14_O_6_	Epicatechin
19	4.45	635.1284	617.1109, 483.1135, 297.0992, 169.0143	C_27_H_24_O_18_	1,2,6-Trigalloyglucose
20	4.5	173.0876	173.0876, 137.0248	C_7_H_14_N_2_O_3_	L-Theanine
21	4.51	457.0849	457.0849, 169.0189, 125.0259	C_22_H_18_O_11_	Epigallocatechin gallate
22	4.78	479.0826	479.0826, 461.0720, 316.0297	C_21_H_20_O_13_	Myricetin 3-glucoside
23	5.2	441.0843	441.0843, 289.0743, 169.0142	C_22_H_18_O_10_	Epicatechin 3-O-gallate
24	5.3	745.1413	745.1413, 441.0834, 303.0519, 169.0149	C_37_H_30_O_17_	Epigallocatechin-(4β→8)-epicatechin-3-O-gallate ester
25	5.8	467.0957	467.0957, 305.0661, 179.0344	C_24_H_20_O_10_	Epigallocatechin-3-O-caffeate
26	6.05	331.065	169.0143, 125.0244	C_13_H_16_O_10_	1-Galloyl-glucose
27	6.08	409.0923	409.0923, 179.0344, 137.0212	C_22_H_18_O_8_	Epicatechin 3-O-p-hydroxybenzoate
28	6.25	315.0505	315.0505, 243.3030, 93.0340	C_16_H_12_O_7_	Pollenitin
29	6.98	303.0442	303.0442, 273.0410, 125.0240	C_15_H_12_O_7_	Dihydroquercetin
30	9.44	353.3087	163.0000, 135.0026	C_16_H_18_O_9_	Chlorogenic acid
31	11.12	781.2169	763.2086, 739.2086, 635.1612, 285.0399	C_35_H_42_O_20_	Kaempferol 3-rhamnosyl-(1->3)(4′′′-acetylrhamnosyl)(1->6)-glucoside
32	12.4	755.4087	755.4087, 737.3458, 575.2341, 285.0399, 163.0670	C_33_H_40_O_20_	Kaempferol 3-(2G-glucosylruntinoside)
33	21.97	353.0878	353.0878, 175.0031	C_20_H_18_O_11_	Noreugenin

## Data Availability

http://www.ncbi.nlm.nih.gov/bioproject/857287 and www.ebi.ac.uk/metabolights/MTBLS5380, both accessed on 1 January 2021.

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
