# Peer review of "Polyphenol-Rich Liupao Tea Extract Prevents High-Fat Diet-Induced MAFLD by Modulating the Gut Microbiota"

_nutrients, 2022, doi:10.3390/nu14224930_

Round 1

Reviewer 1 Report

Polyphenol-rich Liupao tea extract prevents high-fat diet-induced MAFLD by modulating gut microbiota

Tang W, Mengfei Yuan, Zewen Li, et al.

Nutrients-1042765

Overall Impression:  The overall conclusion of this study is that phenol-rich Liupao tea extract antagonizes the ability of a high fat diet to induce MAFLD in ApoE-/- male mice.  The authors present data that suggest the polyphenol extract alters the composition of the composition of the intestinal microflora and this ability in turn blunts the ability of the HFD to cause MAFLD.  This is a data-rich manuscript with perhaps too much data presented in single figures, making it difficult readers and this reviewer to interpret and verify their results.  However I was impressed with the approach, especially the fecal transplants.  Listed below are specific comments that will help improve this manuscript.

Specific Comments:  

1.    This study uses only male mice.  The authors might mention the work of Huang et alPublished online 2021 Dec 15. doi: 10.1136/bmjopen-2021-056260  .  This report shows that MAFLD is more highly associated with males, but in the elderly population females are only slightly less represented in the population of patients with MAFLD

2.   For controls in this study, if the PLE was given by intragastric gavage daily, then the other groups should also receive sterilized pure water without PLE via gastric gavage daily.

3.   Some of the English needs to be corrected.  For example, line 86 states that the outcomes of mice were blindly collected.  While this elicited a chuckle by this native English speaker, it does not belong in a scientific publication. Also, I prefer that term “terminated or killed” to “sacrificed”.  The later implies some type of pagan religious ceremony. 

4.   The amount of data per figure is too much, especially figure one, which makes it difficult for the reader to interpret the data and come to the same of different conclusions than the authors.

5.   The n of different groups various for the different treatments.  How were the number of animals for different treatments determined?  Were power calculations performed?

6.   I commend the authors for the fecal transplantation experiments.  This part of the study is why I rated the significance of the work high significance.

7.   There is duplication in sentence 319 and frankly the English in the section needs to be improved

It would help the readers if the authors would mention what gaps of knowledge need to be determined to explore the potential of PLE as a nutraceutical treatment for MAFLD and also what directions the authors are considering to follow to further explore the mechanisms by which PLE exerts its antagonism of MAFLD.

Author Response

Dear Editor,

Thank you very much for your previous e-mail on November 3rd, 2022 regarding our manuscript ‘‘Polyphenol-rich Liupao tea extract prevents high-fat diet-induced MAFLD by modulating gut microbiota’’ (Manuscript ID: nutrients-1942765). We are very pleased to know that our manuscript is potentially acceptable for publication in the journal, subject to the further revisions suggested by the reviewers. We are very grateful for your substantial and helpful advice regarding our manuscript, and we are pleased to receive the reviewers’ overall positive comments about our work. We thank the three reviewers for their substantial and valuable comments, including their careful reading and checking of the manuscript, which greatly helped us improve the paper. In the revised manuscript, we marked all the changed words, sentences and paragraphs in red text. Our revisions and responses to the editor’s and reviewers’ comments (italic text) are provided below.

To Reviewer 1:

  1. This study uses only male mice. The authors might mention the work of Huang et alPublished online 2021 Dec 15. doi: 1136/bmjopen-2021-056260. This report shows that MAFLD is more highly associated with males, but in the elderly population females are only slightly less represented in the population of patients with MAFLD

Response 1: It is grateful for your professional review towards this article. In the Section 2.3 line 82 of manuscript, Huang et al. was cited as “According to the report from Huang et al [15], MAFLD is more highly associated with males”. Thanks again for your suggestion to make our paper more logical.

  1. For controls in this study, if the PLE was given by intragastric gavage daily, then the other groups should also receive sterilized pure water without PLE via gastric gavage daily.

Response 2: Thanks for your reminder. The mice in both NFD group and HFD group were gavage 200uL sterilized water daily. The description could be found in section 2.3, line 88, and it was amended as “The mice in both normal-fat diet and high-fat diet group were also gavage 200uL sterilized water daily.”

  1. Some of the English needs to be corrected. For example, line 86 states that the outcomes of mice were blindly collected. While this elicited a chuckle by this native English speaker, it does not belong in a scientific publication. Also, I prefer that term “terminated or killed” to “sacrificed”.  The later implies some type of pagan religious ceremony.

Response 3: Thanks for pointing out these ridiculous mistakes. The sentence “Outcomes of mice were blindly collected.” was removed. The word “sacrificed” was substituted with “killed”. For the grammar of this manuscript, a third-party commercial editing company was accepted. And the changes were marked up with red color highlights.

  1. The amount of data per figure is too much, especially figure one, which makes it difficult for the reader to interpret the data and come to the same of different conclusions than the authors.

Response 4: It is appreciated for your careful review of our paper. According to your suggestion, we have reordered the charts according to the sequence in which the charts appear in the article. At the same time, the two pictures “a” and “b” were deleted in Figure 3 since these two figures are not mentioned in the text. In addition, the traces of changes to the chart serial number are also marked in red. Thanks again for your careful review, your suggestions made our manuscript more understandable.

  1. The n of different groups various for the different treatments. How were the number of animals for different treatments determined? Were power calculations performed?

Response 5: In our pre-experiment, compared with the high-fat model group, PLE could significantly reduce the level of NAFLD in the high-fat model group. The NAS score of liver pathological slices that drop by about two-fifths. Therefore, according to the results of the pre-experiment, we conducted a Power calculation for the formal experiment: Compare k Means: 1-Way ANOVA Pairwise, 2-Sided Equality (the relevant website is http://powerandsamplesize.com/Calculators ):The value of Group A is set to 10 according to the actual NAS score of the experiment, while the value of Group B is set to 6, the difference between groups is set to 1.8 according to the pre-experimental value, the Number of Pairwise Comparisons is set to 3, Power, 1−β is set to 0.8, Type I error rate, α is set to 0.5. The minimum sample size of the calculated experimental requirements is 4.

In formal experiments, we have considered the loss rate caused by reasons such as model failure and animal death, usually around 10-20%. Therefore, the four groups of NFD, NFD+PLE, HFD, and HFD+PLE were pre-set with 9 mice in each group for the experiment. The period that the mice were given gavage intervention was 3 months. During these 3 months, two mice in each group of NFD, NFD+PLE and HFD+PLE died unexpectedly due to unexpected reasons such as gavage technique and mice attacking with each other. Therefore, in the follow-up tests: pathological sections of liver and ileum, mRNA level determination of signaling axis gene TLR4-MyD88 in liver tissue, inflammatory factor level determination in blood, and 16S mRNA sequencing analysis of stool, etc., The number of mice in the NFD, NFD+PLE, HFD+PLE groups was 7, and the number of mice in the HFD group was 9. In addition, due to the simultaneous detection of multiple indicators, the samples used for the detection of mouse serum and fecal metabolomics have insufficient individual samples, but in the mouse serum metabolome, the sample quantity is 5-7, while in the mouse fecal metabolome the number of samples is 6 or 7, which could also be able to meet the requirement of statistical difference calculation. Since the qPCR experiment led to the depletion of mouse liver samples, 5 samples in each group for Western blotting Test of liver TLR4 and MyD88 proteins was considered. According to the calculation results of Power calculation, it also met the requirements of statistical validation.

Considering that the number of mice in the HFD and HFD+PLE groups that completed the 3-month intervention were 7 and 9, respectively, in our formal FMT experiment, the number of mice in the two recipient groups was 7, and the mice had no accidental death in the formal experiment. In the subsequent experiments: pathological sections of liver and ileum, mRNA level determination of signaling axis gene TLR4-MyD88 in liver tissue, inflammatory factor level determination in blood, and fecal metabolome experiment, etc. The number of mice in each group of HFD-FMT, HFD+PLE-FMT was 7, and number of samples in the 16S mRNA sequencing analysis of fecal and serum metabolome experiment was 6 for sample consumed. The number of mice in WB experiment was consistent with the donor mice, was 5 per group.

  1. I commend the authors for the fecal transplantation experiments. This part of the study is why I rated the significance of the work high significance.

Response 6: It is appreciated for the compliment and approval from the reviewer. This would encourage us to continue our research in this subdivided area.

  1. There is duplication in sentence 319 and frankly the English in the section needs to be improved

Response 7: Thanks for the carefully review. The redundant part was deleted. The proofreading service of third-party editing company was accepted. And the changes were marked up with red color highlights.

  1. It would help the readers if the authors would mention what gaps of knowledge need to be determined to explore the potential of PLE as a nutraceutical treatment for MAFLD and also what directions the authors are considering to follow to further explore the mechanisms by which PLE exerts its antagonism of MAFLD.

Response 8: Our study found that PLE has great potential in adjunct therapy of MAFLD, and Horizontal-FMT experiments demonstrated that gut microbiota was a key target of PLE to improve MAFLD. Further exploration is needed to determine whether PLE has a regulatory effect on the gut microbiota of NAFLD patients. Secondly, metagenomics investigation is required to determine which core bacterial strains regulated by PLE. Finally, how the bacterial strains protect the intestinal barrier and alleviates NAFLD remains to be further explored. Based on your suggestion, we have added discussion mentioned above and marked red in line 496-501 in the article. We are so grateful for your suggestion that makes the discussion part of our article more dialectical.

Reviewer 2 Report

Dear editor, first of all I thank you for the opportunity to review this article. I believe that the work is exhaustively designed and very well meets the conditions in its present format.

Author Response

It is appreciated for the compliment and approval from the reviewer. This would encourage us to continue our research in this subdivided area.

Reviewer 3 Report

Interesting paper, but the different figures could be improved in their presentation to help the reading of this article.

A short explanation or comparison between MAFLD and NAFLD could be interesting.

Author Response

Dear Editor,

Thank you very much for your previous e-mail on November 3rd, 2022 regarding our manuscript ‘‘Polyphenol-rich Liupao tea extract prevents high-fat diet-induced MAFLD by modulating gut microbiota’’ (Manuscript ID: nutrients-1942765). We are very pleased to know that our manuscript is potentially acceptable for publication in the journal, subject to the further revisions suggested by the reviewers. We are very grateful for your substantial and helpful advice regarding our manuscript, and we are pleased to receive the reviewers’ overall positive comments about our work. We thank the three reviewers for their substantial and valuable comments, including their careful reading and checking of the manuscript, which greatly helped us improve the paper. In the revised manuscript, we marked all the changed words, sentences and paragraphs in red text. Our revisions and responses to the reviewers’ comments (italic text) are provided below.

To Reviewer 3:

1、Interesting paper, but the different figures could be improved in their presentation to help the reading of this article.

Response 1: It is appreciated for your careful review of our paper. According to your suggestion, we have reordered the charts according to the sequence in which the charts appear in the article. At the same time, the two pictures “a” and “b” were deleted in Supplementary Figure 3 since these two figures are not mentioned in the text. In addition, the traces of changes to the chart serial number are also marked in red. Thanks again for your careful review, your suggestions made our manuscript more understandable.

2、A short explanation or comparison between MAFLD and NAFLD could be interesting.

Response 2: It is grateful for your professional review towards this article.  A description of “The prevalence of metabolic-associated fatty liver disease (MAFLD) is increasing worldwide, which is used to be known as non-alcoholic fatty liver disease fatty liver disease (NAFLD)” was added into Introduction of manuscript. The description could find in line30. Thanks again for your careful review, your suggestions made our manuscript more understandable.
